# Smooth Complete Coverage Trajectory Planning Algorithm for a Nonholonomic Robot

**DOI:** 10.3390/s22239269

**Published:** 2022-11-28

**Authors:** Ana Šelek, Marija Seder, Mišel Brezak, Ivan Petrović

**Affiliations:** Laboratory for Autonomous Systems and Mobile Robotics (LAMOR), Faculty of Electrical Engineering and Computing, University of Zagreb, 10000 Zagreb, Croatia

**Keywords:** mobile robot, complete coverage, path planning, path smoothing, velocity profile optimization

## Abstract

The complete coverage path planning is a process of finding a path which ensures that a mobile robot completely covers the entire environment while following the planned path. In this paper, we propose a complete coverage path planning algorithm that generates smooth complete coverage paths based on clothoids that allow a nonholonomic mobile robot to move in optimal time while following the path. This algorithm greatly reduces coverage time, the path length, and overlap area, and increases the coverage rate compared to the state-of-the-art complete coverage algorithms, which is verified by simulation. Furthermore, the proposed algorithm is suitable for real-time operation due to its computational simplicity and allows path replanning in case the robot encounters unknown obstacles. The efficiency of the proposed algorithm is validated by experimental results on the Pioneer 3DX mobile robot.

## 1. Introduction

The task of a Complete Coverage Path Planning (CCPP) algorithm is to generate such a path for a mobile robot that ensures that the robot completely covers the entire environment while following the planned path. There are many real-world applications that require a CCPP algorithm, such as floor cleaning [1,2,3], demining [4,5], automated harvesting [6], lawn mowing [7], autonomous underwater exploration [8,9], etc. To achieve efficient coverage in these applications, the planned path must satisfy some requirements, where the most important ones are maximizing the coverage rate, minimizing the complete coverage time, path length, overlap area, and energy consumption of the robot, and rapid path replanning if the environment changes during the execution of the coverage task. Many CCPP algorithms have been developed, but none of them meet all the above requirements, so there is still room for improvement.

In this paper, we propose a CCPP algorithm which generates smooth complete coverage paths that allow nonholonomic mobile robots to move in optimal time while following them (we called it SCCPP). Moreover, the proposed SCCPP algorithm is suitable for real-time operation due to its computational simplicity and allows path replanning in case the robot encounters unknown obstacles. The existing coverage algorithms in the literature are either non-smooth so they have increased coverage redundancy due to the non-ideal path following, or they have slow path planning and replanning. The SCCPP algorithm combines two of our previous works: the fast coverage planning algorithm [10] with the fast clothoid calculation [11]. The first algorithm is our replanning spanning tree coverage (RSTC) algorithm that generates a path in a low-resolution occupancy grid map to reduce the computational complexity and minimize the overlap rate. The path is the shortest possible coverage path in the corresponding graph, which contains sharp 90° turns. To avoid stopping and rotating the robot at turning points, we used clothoids to smooth the path generated by the RSTC algorithm. The main advantage of the clothoids is the linear change in their curvature, which allows the nonholonomic robot to move in a time-optimal manner while following such a smoothed path.

The SCCPP is the real-time traversable collision-free complete coverage path planning algorithm based on clothoids, which gives minimal path length, the coverage time, and overlap area and maximal coverage rate compared to the state-of-the-art coverage algorithms. Such a path is suitable and feasible for nonholonomic mobile robots.

The remaining of the paper is structured as follows. Section 2 presents the overview of the complete coverage path planning algorithms. The proposed smooth complete coverage path planning algorithm is described in Section 3. The simulation and experimental results are given in Section 4 and Section 5, respectively, followed by the conclusion of the paper.

## 2. Related Work

One of the earliest works on complete coverage path planning is presented in [12], which defines requirements for covering a continuous two-dimensional (2D) environment. Typically, methods for CCPP tasks assume that the environment is known, including the obstacle configuration. A number of methods perform an exact cell decomposition of the environment, e.g., [13,14,15], where free space is decomposed into trapezoidal, rectangular, or triangular cells. The path that completely covers each cell (regarding their shape) can lead to increased coverage redundancy and is time-consuming when the environment changes. Simpler and faster are methods that perform an approximate cell decomposition where the environment is represented with a 2D occupancy grid map, i.e., equally distributed square grid of discrete cells [16,17,18,19].

The size of the square grid cells directly affects the replanning rate and the coverage rate. These two requirements are opposite, i.e., larger cell sizes allow real-time replanning due to lower computational complexity, while smaller ones ensure higher coverage rate at the cost of higher computational complexity. Fast replanning algorithms usually select the cell size equal to the footprint of the robot, while the cell size is much smaller in the algorithms that ensure a high coverage rate, usually from 2 to 10 cm for the cell side [20,21].

A graph can be constructed from the occupancy grid map, where the grid cells are the nodes and the connections between adjacent grid cells are the edges. An optimal CCPP would ensure that the robot completely covers the entire environment by visiting all nodes in the graph only once, but this is a NP -hard problem, known as the Traveling Salesman Problem (TSP) [22]. Therefore, CCPP algorithms use approximate or heuristic solutions (see Figure 1). The complete coverage path planning algorithm based on a numeric potential field approach called the path transform (PT) can be used for the TSP solution [16]. Ant colony optimization (ACO) [23,24] is based on artificial ants colony that are able to generate the shortest feasible path of the TSP successively. Another solution for TSP can be determined by using the Hopfield neural network (HNN) for an optimization that consists of a single layer containing one or more fully connected recurrent neurons [25,26,27]. The Spanning Tree Coverage (STC) algorithm [28] and the Replanning Spanning Tree Coverage (RSTC) algorithm [10] produce the optimal coverage path in linear time. The CCPP algorithms on a high-resolution occupancy grid map have increased coverage rate.

The Complete Coverage D* (CCD*) algorithm [2] has a high coverage rate, which leads to increased coverage redundancy when the environment changes. The hex-decomposition-based coverage planning (HDCP) algorithm [29] for unknown environments plans smooth paths for Dubins vehicles to follow at constant velocity in real-time. Adaptive coverage path planning [30] is aiming to achieve complete coverage with minimal path length and it is efficient in dynamic environments. Complete coverage path planning based on biologically inspired neural network [31] plans collision-free trajectory for real-time coverage task in dynamic environments. The energy constrained online coverage path planning [32] is based on contour following, which causes sharp turns in rectangular environments. The drawbacks of the above algorithms are that the coverage redundancy is increased when dynamic obstacles are present, and a trade-off exists between a higher coverage rate and higher coverage redundancy. Moreover, the energy consumption and the coverage time are high because most of these algorithms generate the path with straight lines that form sharp turns. When the nonholonomic robot follows such a path, it must stop and reorient itself so that the heading of the robot matches the path direction [33]. Authors in [34] do a complete 2D sweep coverage, but their approach does not produce smooth paths, is not computationally efficient, and does not consider dynamic obstacles but only static environments.

To provide optimal and feasible paths with curvature continuity that are easy to follow by nonholonomic mobile robots, path smoothing algorithms are used. A smoothing algorithm provides motion continuity and reduces the execution time of coverage tasks. The first research on finding the shortest curvature constrained smooth paths consisting of straight lines and arcs was done by Dubin in [35]. The presence of discontinuities at the intersection of the straight lines and circular arcs makes them infeasible for real applications. Backman et al. [36] presented a curvature continuous CCPP algorithm using the Dubin curve for agricultural vehicles. The generated path is easy to follow for real-world applications and the algorithm is efficient in coverage time but computationally complex. Yu et al. [37] proposed Dubin’s vehicle-based coverage algorithm, which minimizes the non-working distance in CCPP, reduces the number of turns of the vehicle, maximizes the coverage area, but increases the overlap area. Jin and Tang [38] developed a coverage path planning algorithm which respects the kinematic constraints of the robot, but it is specialized for agricultural or similar purposes and is computationally complex. Lee et al. [39] presented a smooth CCPP algorithm based on Bézier curve, wall following, and high-resolution grid map representation. The algorithm maximizes coverage rate and minimizes energy consumption, but has increased coverage redundancy. All the above smoothing algorithms are not suitable for changing environments due to high computational complexity.

## 3. The Proposed Smooth Complete Coverage Path Planning Algorithm

The pipeline of the SCCPP algorithm, shown in Figure 2, consists of four sequentially computed modules. The first module is the replanning spanning tree coverage algorithm (RSTC), which generates a path composed of straight-line path segments perpendicular to each other (Section 3.1). The algorithm initially requires the map of the environment and the starting pose of the robot. The initial map can be a CAD map or a map created with a simultaneous localization and mapping algorithm (SLAM). The second module is the path smoothing algorithm, which smooths the path generated by the RSTC algorithm using clothoids (Section 3.2). The third module is the velocity profile optimization algorithm, which computes the highest admissible velocity profile along the smoothed path accounting for kinematic and dynamic constraints of the robot (Section 3.3). The fourth module is the trajectory tracking algorithm, which ensures that the robot tracks the optimal 5D trajectory generated by the third module (Section 3.3).

### 3.1. The Replanning Spanning Tree Coverage Algorithm

We use the replanning spanning tree coverage (RSTC) algorithm [10], which we briefly describe here. A map of the environment, given in the Portable Network Graphics (PNG) format, was edited to align the walls with the grid for better performance of the coverage algorithm and downsampled to a low-resolution occupancy grid with equally sized square cells (Figure 3a). Each cell contains occupancy information depending on the obstacle position. Only completely free cells were considered in the path planning and they are divided into four equal subcells of size *D*, where *D* is the diameter of the circumscribed circle around the robot’s footprint (Figure 3b).

To create an optimal path, which visits each subcell exactly once, a spanning tree is constructed (Figure 3c). Free 2D-size cells induce an undirected graph structure whose nodes are the centers of the cells, while the edges are the connecting lines between centers of the adjacent cells that share a common cell side. A spanning tree of an undirected graph is a subgraph that includes all nodes of the graph, with a minimum possible number of edges. A spanning tree is found in linear time by using the Breadth First Search (BFS) algorithm, meaning that the candidate neighbor nodes are kept in a queue. The pseudocode for spanning tree determination is given by Algorithm 1. The construction of the spanning tree starts from the node which contains the robot’s position. All its neighbor nodes are inserted into the queue. The second node of the tree defines the first edge of the spanning tree, which has a direction aligned with the robot’s orientation, if possible. The spanning tree continues to grow by removing from the queue and connecting to the last added node of the tree the node that is its neighbor. If this node has no neighbors in the queue, the node added one step earlier is compared to the queue, and so on. The tree is constructed when there are no nodes in the queue.
**Algorithm** **1** Pseudocode for the spanning tree1:**Input:** occupancyGridMap, currentPose2:**Output:** spanningTree3:insert starting cell which contains the robot’s position in the queue *spanningTree*4:**while** all free cells in occupancyGridMap have been visited **do**5:    determine all orthogonal and unvisited neighbors of the current cell moving counterclockwise and add them to the queue neighbor6:    **for** all cells in queue neighbor **do**7:        **if** distance of current cell in *spanningTree* and neighbor ≤1 **then**8:           add current cell to the queue *spanningTree*9:           set current cell in occupancyGridMap as visited10:         go to the neighbor cell11:      **end if**12:    **end for**13:    **if** distance of current cell in *spanningTree* and neighbor >1 **then**14:        **for** cells in queue spanningTree **do**15:           **if** distance of current cell in *spanningTree* and neighbor ≤1 **then**16:               go to the neighbor cell17:           **else**18:               move to the previous cell in queue spanningTree19:           **end if**20:        **end for**21:    **end if**22:**end while**23:return the spanning tree

Once the spanning tree is created, the coverage path computation begins. The pseudocode for the path planning is given by Algorithm 2. The robot is assumed to follow the path around the spanning tree, always on the right side, until it completely covers all subcells. The coverage ends when the robot gets to the start subcell. The result is the complete coverage path, which consists of a series of connected lines (Figure 3d).
**Algorithm** **2** Pseudocode for the path planning1:**Input:** spanningTree, currentPose2:**Output:** RSTCpath3:**while** it reaches the starting subcell again **do**4:    calculate the direction of the spanning tree form current cell to the next first neighbor which is connected with the edge in the spannning tree5:    mark current subcell as visited6:    add subcell center coordinates in the queue RSTCpath7:    go to neighbor cell8:    **if** dynamic obstacle is detected **then**9:        end the path planning algorithm10:   **end if**11:**end while**12:return the RSTC path

Figure 1 shows a comparison of the RSTC algorithm with heuristic methods that solve TSP: the PT, ACO, and HNN algorithms. The length of the path for all methods is 76 m. The RSTC path has the smallest number of turning points (42 turning points compared to 48 for PT, 56 for ACO, and 50 for HNN). The higher number of turns results with longer coverage execution time. Therefore, our algorithm produces the coverage path more suitable for nonholonomic robots than other TSP solutions. The RSTC path has the lowest computational time (measured in Matlab/Python implementation) (0.32 s compared to 1.56 s for PT, 124 s for ACO, and 0.6 s for HNN).

While the robot follows the planned path and visits subcells one by one, it may encounter an unknown obstacle (Algorithm 3). In such a case, the new spanning tree is created for the rest of the unvisited grid cells and the path is recomputed. The robot continues to follow the new path from the right side of the spanning tree until it returns to the cell where the replanning started. It then continues to follow the previously planned path. If all cells are visited then the shortest path around the obstacle is determined and connected with the previously planned path. If the unknown obstacles free occupied cells, set these cells as free in the occupancy grid map. Add these cells to the previously determined spanning tree if the current cell is the diagonal neighbor of the occupied cells. The new path around this spanning tree is determined. The complete coverage algorithm ends when the robot returns to the start subcell of the initial path. Since only completely free cells are considered, the coverage path is optimal and minimizes the overlap area.

In summary, the main advantages of the RSTC algorithm are that it generates the shortest path, minimizes the overlap area, and allows real-time path replanning in changing environments. However, its drawbacks are sharp turns of the planned path where a robot has to stop and reorient itself to continue, which is inefficient regarding the task duration and energy consumption.
**Algorithm** **3** Pseudocode for the complete coverage path planning1:**Input:** Map, currentPose2:**Output:** RSTCpath3:determine occupancy grid map based on png map image4:determine spanning tree based on Algorithm 15:determine the RSTC path based on Algorithm 26:**while** it reaches the starting subcell again **do**7:    **if** unknown obstacle is detected **then**8:        occupy cells in which unknown obstacle is detected9:        **if** exist the unvisited grid cells **then**10:           determine the new spanning tree based on Algorithm 1 for the rest unvisited grid cells11:           determine the new RSTC path based on Algorithm 2 for the rest unvisited grid cells12:       **else**13:           determine the RSTC path around the obstacle so that the minimum number of double-covered subcells is obtained and connect it with previously planned path14:       **end if**15:   **end if**16:   **if** unknown obstacle cleared occupied cells **then**17:        set these cells as free in the occupancy grid map18:        **if** current cell is diagonal neighbor of the occupied cell **then**19:           add these cells in the previously determined spanning tree20:           determine the new RSTC path based on Algorithm 221:        **end if**22:    **end if**23:**end while**24:return the RSTC path

### 3.2. Path Smoothing

The task of the path smoothing algorithm is to smooth the path generated by the RSTC algorithm at the sharp turns to allow continuous motion of the robot without stopping. The path smoothing algorithm [11] is used, which uses clothoids to smoothly connect adjacent lines of the path.

The coordinates of a general clothoid are:
(1a)$$x\left(s\right)=x_{0}+\int_{0}^{s}\cos(\theta_{0}+\kappa_{0}\xi +\frac{1}{2}c\xi^{2})d\xi ,$$
(1b)$$y\left(s\right)=y_{0}+\int_{0}^{s}\sin(\theta_{0}+\kappa_{0}\xi +\frac{1}{2}c\xi^{2})d\xi ,$$
where $\left(x_{0},y_{0},\theta_{0}\right)$ is the initial robot pose, the parameter $\kappa_{0}$ is the initial curvature at s≥0, which denotes the arc length, and the parameter *c* is the sharpness, which describes how much the curvature changes with traveled distance. Curvature κ can be expressed as a linear function of the arc length *s*:(2)$$\kappa \left(s\right)=\kappa_{0}+cs.$$

The Equation (1) contain Fresnel integrals, which are transcendental functions that cannot be solved analytically, making them difficult to use in real-time applications. To enable the use of clothoids in real-time, various methods have been developed and we use the one that is particularly fast and thus very suitable for real-time applications [11]. It is based on a so-called basic clothoid with bounded approximation error [40] whose coordinates can be stored in a lookup table, and the coordinates of any other clothoid can be determined based on the points of the basic clothoid by rescaling, rotating, and translating.

An example of the smoothed sharp turn is shown in Figure 4, where the points *S*, *T*, and *G* denote the start, sharp turn, and the goal position, respectively. The smooth path is shown as a red curve consisting of four parts: a straight line $\bar{SC_{1}}$, the first clothoid from $C_{1}$ to $C_{2}$, the second clothoid from $C_{2}$ to $C_{3}$, and a straight line $\bar{C_{3}G}$. The two clothoids are symmetric–the second one is obtained by mirroring the first one at the bisector of the angle spanned by the lines $\bar{ST}$ and $\bar{TG}$.

The parameter *e* is the deviation from the planned path. If it is high, the path curvature is low and the robot can drive at a higher velocity. Since for our problem, each sharp turn is at a $90^{\circ }$ angle, the same deviation parameter *e*, and consequently the same pair of symmetric clothoids can be applied for all of them, which additionally simplifies the smoothing.

We determined the maximum deviation $e_{\max}$ to ensure that the robot’s footprint does not collide with the obstacle cell at the point of maximal deviation, i.e.:(3)$$e_{\max}\leq \frac{D\sqrt{2}-D}{2},$$
where $D\sqrt{2}$ is the subcell diagonal, and *D* is the diameter of the circumscribed circle around the robot’s footprint. In our setup it is D=0.5 m, so we choose $e_{\max}=0.1$ m. From $e_{\max}$ it is possible to determine the maximum curvature in point $C_{2}$ (see [11]), so we obtain $\kappa_{\max}=6.4$
$m^{-1}$.

From the radial acceleration, which is $a_{\mathrm{radial}}=v\times \omega =v^{2}\kappa$, where *v* and ω are the linear and angular velocity of the robot, respectively, we can get the velocity limit at the curvature extrema as:(4)$$v_{\min}=\sqrt{\frac{a_{{\mathrm{radial}}_{\max}}}{\kappa_{\max}}}.$$

Since $a_{{\mathrm{radial}}_{\max}}$ is characteristic of the mobile robot, and in our setup it is $a_{{\mathrm{radial}}_{\max}}=0.1$
${\mathrm{m/s}}^{2}$, this gives the velocity limit $v_{\min}=0.125$ m/s, which is used later in the velocity profile optimization module.

Figure 5 shows a part of the original and smoothed path with two positions of the robot. It can be noticed that the robot’s circular footprint when moving along the smoothed path is always within the free subcells of size *D*. It does not collide with the obstacle cell even at points where a deviation of the smoothed path from the original path is the largest (noted with *e*).

As described in Section 3.1, the RSTC algorithm recalculates the path for the unvisited grid subcells when the robot encounters an unknown obstacle in the neighboring cells. Due to the computational simplicity of the proposed path smoothing algorithm, the smoothing of the replanned path can be conducted online so that the robot does not need to slow down or stop.

### 3.3. Velocity Profile Optimization and Trajectory Tracking

Although the path smoothing algorithm takes into account the kinematic and dynamic constraints of the robot, it only outputs *x* and *y* coordinates and curvature of the smooth complete coverage path. To ensure time-optimal robot motion along the smooth path, the velocity profile with maximum allowable velocities should be calculated. The turning points of the smoothed path (e.g., as point $C_{2}$ in Figure 4) are the local extrema points of the smooth path curvature, i.e., at these points, the turning radius reaches a local minimum, and consequently, the velocity is locally lowest. The maximum allowable velocity of the robot at a turning point is determined by the radial acceleration limit $a_{\mathrm{radial}}$. The robot must decelerate and accelerate before and after the turning point as much as maximally tangentially allowed by the acceleration limit. The robot should always remain within its acceleration limits. The overall acceleration is calculated as:(5)$$a=\sqrt{a_{\mathrm{radial}}^{2}+a_{\mathrm{trans}}^{2}}$$
where $a_{\mathrm{radial}}=v^{2}\kappa$ is the radial acceleration of the robot and $a_{\mathrm{trans}}=\frac{dv}{dt}$ is the translation acceleration.

We used the velocity profile optimization algorithm described in detail in [41]. It outputs the optimal linear and angular velocities of the robot based on the points of the smooth complete coverage path and the driving limitations. Linear and angular velocities, together with the robot position and orientation, result in the 5D trajectory ${\left(x,y,\theta ,v,\omega \right)}^{T}$. This trajectory is the input to the trajectory tracking algorithm based on the Kanayama controller [42]. This controller ensures that the robot tracks the optimal 5D trajectory by dumping perturbations of the robot pose provided by the localization module and velocity measurements from the wheel encoders.

The kinematic model of the differential drive mobile robot can be represented as follows:(6)$$\left[\begin{array}{c} \dot{x}\left(t\right)\\ \dot{y}\left(t\right)\\ \dot{\theta }\left(t\right) \end{array}\right]=\left[\begin{array}{cc} cos\theta (t) & 0\\ sin\theta (t) & 0\\ 0 & 1 \end{array}\right]\left[\begin{array}{c} v(t)\\ \omega (t) \end{array}\right],$$
where *t* is time, (x(t), y(t)) is position, θ(t) is orientation, v(t) is linear velocity, and ω(t) is the angular velocity of the mobile robot. For the system (Equation 6), the nonholonomic constraint is:(7)$$-\dot{x}\left(t\right)\cdot sin\theta \left(t\right)+\dot{y}\left(t\right)\cdot cos\theta \left(t\right)=0,$$
which constrains the drive wheels to roll and prevents slippage. A path is feasible for nonholonomic mobile robot if it can follow the path with the reference velocity commands.

The main task of the tracking control for mobile robots is to find appropriate velocities v(t) and ω(t) to achieve the control objective. Inputs for the Kanayama controller are the reference configuration $p_{r}$$={\left(x_{r},y_{r},\theta_{r}\right)}^{T}$, which is taken from the 5D trajectory at the current time instant, and the measured current configuration $p_{c}={\left(x_{c},y_{c},\theta_{c}\right)}^{T}$. The Kanayama controller outputs are the corrected reference linear and angular velocities. The first step of the control law is to compute the error configuration $p_{e}$ as the difference between $p_{r}$ and $p_{c}$, which must converge to zero. For the kinematic model given by (Equation 6) and (Equation 7), the expression for the error configuration is:(8)$$p_{e}=\left[\begin{array}{c} x_{e}\\ y_{e}\\ \theta_{e} \end{array}\right]=\left[\begin{array}{ccc} cos\theta_{c} & sin\theta_{c} & 0\\ -sin\theta_{c} & cos\theta_{c} & 0\\ 0 & 0 & 1 \end{array}\right]\left(p_{r}-p_{c}\right).$$

Using the error configuration $p_{e}$, the reference linear and angular velocities $v_{r}$ and $\omega_{r}$, the corrected linear and angular velocities *v* and ω can be calculated as follows:(9)$$\left[\begin{array}{c} v\\ w \end{array}\right]=\left[\begin{array}{c} v_{r}cos\theta_{e}+K_{x}x_{e}\\ \omega_{r}+v_{r}\left(K_{y}y_{e}+K_{\theta }sin\theta_{e}\right) \end{array}\right],$$
where $K_{x}$, $K_{y}$ and $K_{\theta }$ are the positive constant parameters of the controller.

Figure 6 shows the smoothed path and the tracked trajectory for the example shown in Figure 4. There is no significant deviation of the tracked trajectory from the smoothed path and the mobile robot moves smoothly from one segment of the path to another.

Figure 7 shows linear and angular velocities for the example shown in Figure 4 calculated by the velocity profile optimization procedure (blue line) and the real linear and angular velocity profile, while the robot tracked the trajectory (magenta diamonds). The corrected reference velocities calculated by Kanayama’s algorithm are shown with black diamonds. The robot in the simulation reached exactly the corrected reference velocity with a delay of one control time step.

## 4. Simulation Results

We tested the proposed motion planning approach in three scenarios: the *Lab scenario* (Section 4.1), *Aula scenario* (Section 4.2), and *Gallery scenario* (Section 4.3). For each scenario, the proposed SCCPP algorithm is compared with the original RSTC algorithm without smoothing (called the CCPP algorithm in the following), and the results are discussed in Section 4.4. We also compared SCCPP with CCD* and HDCP algorithms for the *Gallery scenario*. CCD* was selected for comparison because it is a graph search method based on D* search of the high resolution grid map of the environment. HDCP was selected for comparison because it plans smooth paths based on Dubins curve, which is followed at constant velocity in real-time.

We used a receding horizon control (RHC) algorithm developed within our research group [43] for the path following of the the CCPP without smoothing. This algorithm can adapt to dynamic changes in the environment. If the selected subgoal is too close to the detected obstacle, a new subgoal is chosen in the local vicinity from the critical one. With this procedure, the robot follows the planned path with a minimal drift.

The simulations were performed on an Asus ROG Strix SCAR II (Intel i7-8750H, 16 GB DDR4 RAM). The algorithms were implemented and tested in the Robot Operating System (ROS). The stage simulator was used for the simulations.

To create the environment map, for each test scenario the environment was explored through sensors on the Pioneer 3DX robot and data was collected by the *slam_gmapping* ROS package. Then, the map was created and post-processed to align walls to the grid for better performance of the coverage algorithm. The post-processed map is resampled to an occupancy grid map with low resolution. The size of the cells is 1×1 m and each cell is divided into four 0.5×0.5 m sub-cells, which are approximately of the same size as the robot’s footprint. The *amcl* ROS package was used for the robot localization in created maps.

The *Lab scenario* tests how the algorithm works in a small room with static and dynamic obstacles. The *Aula scenario* tests how the algorithm works in a large hallways with a large number of static obstacles. The third scenario is the *Gallery scenario*, which tests the algorithm performances in narrow spaces. To validate the performances of the SCCPP algorithm and to compare it with other algorithms, the path length and execution time were measured, and the coverage rate and coverage redundancy were calculated.

The coverage rate is calculated as:(10)$$\mathrm{Coverage}\;\mathrm{rate}=A_{p}/\left(A_{t}-A_{o}\right)\cdot 100\%,$$
where $A_{t}$ is the total area of the environment, $A_{o}$ is the area of obstacles, and $A_{p}$ is the area covered by the driven trajectory of the robot. The coverage redundancy is calculated as follows:(11)$$\mathrm{Coverage}\;\mathrm{redundancy}=A_{r}/A_{p}\cdot 100\%,$$
where $A_{r}$ is the overlap area, calculated as the sum of all subcells visited more than once.

In all figures below that show the paths or trajectories, static obstacles are shown with green dots, and if they partially occupy a cell, the entire cell is shown as occupied (gray cells), the spanning tree is shown as a blue line, the RSTC path as a black line, the smoothed RSTC path as a red line, and the tracked trajectory as a green line.

### 4.1. The Lab Scenario

The coverage starts at cell (7, 4) which is the starting cell for the spanning tree construction and the path circumnavigates around the constructed spanning tree (see Figure 8a). The robot follows the path from the right side of the spanning tree and covers each subcell exactly once. The coverage is complete when the robot returns to the start subcell.

The path is created with straight lines that form sharp turns. In each turn for the CCPP variant, the robot must stop and reorient itself so that the robot heading is equal to the path direction. For this reason, the linear velocity is zero and the angular velocity is close to the maximal value. On straight sections of the path, the linear velocity is maximal and the angular velocity is zero. Figure 8c shows the calculated linear and angular velocities (blue line) and the real linear and angular velocities of the robot executed by the motors (red line).

To improve the coverage and reduce the execution time, the smoothed variant–the SCCPP algorithm is used. The output of this algorithm is the smoothed path that circumnavigates around the constructed spanning tree (see Figure 8b). In this way, sharp turns are now smoothed and the robot can travel without stopping.

The execution of the SCCPP algorithm can be examined from the linear and angular velocities shown in Figure 8d. The blue lines are the linear and angular velocities calculated by the optimization process, and the red lines show how the linear and angular velocities changed as the robot tracked the trajectory. It can be observed that the linear velocity is not zero while the robot has to overcome the turn as opposed to zero velocities in Figure 8c, where the smoothing algorithm is not used.

The replanning SCCPP algorithm is executed in a dynamic environment. Dynamic changes can be detected by the robot in neighboring cells of the current cell where the robot is currently located; see Figure 9a for an example where the obstacle is detected in cell (4, 4). All unvisited cells are considered as nodes for the new spanning tree calculation and the visited subcells are not part of the new path coverage, see Figure 9b. After the new spanning tree is determined for the rest of the unvisited grid cells, the new path is determined. The path always follows the spanning tree counterclockwise. When the robot returns to the cell where the replanning started, it continues to follow the path from the right side of the spanning tree that was already formed during the previous path planning. Therefore, this part of the spanning tree is not shown in Figure 9b from cell (4, 4) to starting cell (7, 5). The robot traverses these visited grid cells by visiting only subcells that are not yet visited. The complete coverage algorithm ends when the robot gets to the starting subcell. The red dots in Figure 9 are the position of the robot where the replanning algorithm is executed. In this example, the replanning process is fast enough that the robot does not need to stop for the recalculation of the smoothed path. The recalculation of the smoothed RSTC path takes less than 1 ms on average.

### 4.2. The Aula Scenario

Similar behavior of the CCPP and SCCPP algorithms can be observed in the example with few hallways and more static obstacles; see Figure 10. Static obstacles are shown with green dots and if they partially or fully occupy a cell, the cell is presented as occupied (gray cells). Some static obstacles are detected by the laser outside the building boundaries because there are windows in parts of the hallway and these grid cells are represented as occupied. The direction of the first segment of the complete coverage path is −90° due to the initial orientation of the robot. The coverage starts at cell (6, 32), which is the starting cell for the spanning tree construction and the path circumnavigates around the constructed spanning tree, but this is not represented in Figure 10. Due to the transparency of the figure we presented, only a smoothed path (red line) and driven trajectory (green line). The smoothed path does not contain sharp turns and the robot can drive the trajectory without stopping and reorienting itself to align heading with the path direction. This improves the coverage rate and reduces the execution time of the coverage task. The SCCPP variant outperforms the CCPP algorithm here as well.

### 4.3. The Gallery Scenario

This scenario has the most turns due to the narrow dimensions of the environment. The starting cell for the spanning tree construction and the RSTC path is (11, 3), see Figure 11. Again, the SCCPP algorithm variant is faster than the CCPP variant.

The SCCPP algorithm is compared to the CCD* and HDCP algorithms in this scenario. The complete coverage path of the CCD* algorihm is shown in Figure 12 with the black line, and visited cells are marked by different colors according to the number of visits, so some parts of the map are visited nine times.

The complete coverage path of the HDCP algorithm is shown in Figure 13 with the green line. Partially occupied cells are shown in gray hexagons and only completely free hexagonal cells are used for the coverage task. The path begins in the cell (11, 3) and ends in the cell (6, 2). Some cells are visited more than once due to the narrow environment and low resolution occupancy grid map.

### 4.4. Discussion

The results of the CCPP and SCCPP comparison in all three scenarios are given in Table 1. When the smoothing algorithm is used, the length of the tracked trajectory is shorter and the execution time is reduced compared to the results obtained without smoothing. In the *Gallery scenario*, the RSTC algorithm provides a path with many sharp turns and because of the discontinuity during the motion, the execution time for the complete coverage is about 10% longer for CCPP than for SCCPP. It can be noted that the environment configuration and the position of the obstacles can affect the execution time and the total length of the tracked trajectory. In addition, the CCPP algorithm has higher localization errors at points where the robot rotates in place. These cause higher uncertainty in the robot’s pose and the deviation from the tracked trajectory is larger than in the SCCPP examples. This also leads to an increase in the coverage redundancy.

From these three scenarios, it can be observed that the SCCPP algorithm has, on average, a 6.5% shorter tracked trajectory, an 8.8% reduction in coverage execution time, a 4.5% better coverage rate, and an 82.34% lower coverage redundancy than the original CCPP algorithm. The better coverage rate of the SCCPP compared to the CCPP can be explained by higher path following accuracy, while accuracy of the CCPP is worsened by sharp turns. To provide the scalability of the proposed algorithm, each scenario has a different size of the environment, which leads to different number of nodes (cells) in a spanning tree construction. Because the number of nodes is proportional to the computation time, *Lab scenario* has the smallest and *Aula scenario* has the biggest elapsed time required to calculate the complete coverage path. For each scenario, SCCPP has a larger time than CCPP for the time required to calculate the smooth path. The time increases linearly with the number of nodes, while the number of nodes increases quadratically with the map dimensions.

The SCCPP algorithm is compared to the CCD* and HDCP algorithms, and the results are shown in Table 2. Compared to CCD* and HDCP, SCCPP has the shortest coverage path length (42% shorter than CCD* and 21% shorter than HDCP), the shortest coverage time (47% shorter than CCD* and 53% shorter than HDCP), and the smallest coverage redundancy (82% smaller than CCD* and 40% smaller than HDCP). However, SCCPP has worse coverage rate compared to CCD* (27% lower), and better coverage rate compared to HDCP (15% higher). The reason of the lower coverage rate of both the SCCPP and HDCP algorithm is the used low resolution occupancy grid map. However, the high resolution occupancy grid map results with higher coverage redundancy in CCD*.

The coverage rate for the SCCPP algorithm can be increased if a wall following method is used, but this also increases the redundancy. The wall following algorithm used after SCCPP is presented in Figure 14 with driven trajectory (red line) and robot’s footprint at every point on the trajectory (degradation from black to gray). The coverage rate is increased to 97.64% with the use of the wall following, and the coverage redundancy is increased to 24.64%.

The advantages of our SCCPP algorithm are completeness of coverage, robustness to environmental shape and initial robot pose, optimal path that visits all subcells exactly once, time efficiency, low coverage redundancy, and fast replanning. The coverage rate can be significantly increased if the wall following method is used. The limitation is that the algorithm requires a priori knowledge about the workspace. The execution of the proposed smooth complete coverage path planning algorithm is around 10 ms and it is suitable for real-time operation due to its computational simplicity.

## 5. Experiments on a Real Robot

The experiment (the experiments are demonstrated in the accompanying video available here: https://youtu.be/VEAGHIAIpRA accessed on 18 October 2022) was performed on the Pioneer 3DX robot with a SICK laser sensor LMS200. To have a similar scenario as in the simulation, the experiment was performed in the *Lab scenario*, where a new map had to be created due to some minor changes in the placement of the furniture. First, the *Lab* was explored using the laser sensor on the robot, and the data of the environment was collected using the *slam_gmapping* ROS package. After the environment was explored, the map was created and the package *amcl* ROS was used for robot localization in the created map.

The SCCPP algorithm was executed and the results are shown in Figure 15. The static obstacle configuration is represented as green dots, which are shown as occupied cells (gray cells in Figure 15) in the occupancy grid map. The initial position of the robot (x=5.25 [m], y=2.75 [m]) is marked with a blue star and this is the starting position for the coverage of the environment. The blue line is the spanning tree determined by RSTC algorithm, the red line is the smoothed RSTC path, and the green line is the driven trajectory by the Pioneer 3DX robot.

There are some deviations between the smoothed path determined by the SCCPP algorithm (red line) and the real trajectory tracked by the robot (green line), especially when dealing with parts of the path that are curved. This problem is due to the inaccurate and noisy localization of the robot. Another problem is the hardware setup. We used serial communication between the robot and the laptop with ROS, which caused a delay of three cycles in sending the calculated velocities to the robot. In Figure 16, the calculated linear and angular reference velocity profile are shown with blue lines and the actual linear and angular velocity profile of the robot during trajectory tracking is shown with magenta lines. For the experiment with the real robot, we used 0.5 m/s and 0.75 rad/s as the maximal linear and angular velocities, respectively. The delay of three cycles caused the corrected reference velocity by the Kanayama tracking algorithm to deviate from the optimized velocity profile of the smoothed path. This is shown in Figure 17, where the first 1.5 s of the linear velocity profile from Figure 16 are shown magnified.

Table 3 presents the experimental evaluation of the SCCPP algorithm efficiency on the Pioneer 3DX robot for the *Lab scenario*. The coverage path length is 53.85 m, the coverage execution time is 179.5 s, the coverage rate is 77.7%, and the coverage redundancy is 12.96%. The trajectory tracking error is determined as the difference of the calculated trajectory (Figure 15 red dotted line) and tracked trajectory (Figure 15 green line) and it is 6.43 $m^{2}$. When compared with simulation results in Table 1 for the *Lab scenario*, although the maps are slightly different, it can be observed that the coverage rate in both cases is approximately equal, which confirms that our algorithm is equally efficient in the real robot setup. The coverage redundancy is slightly worse on a real robot, which is mainly caused by noisy localization and delay in the control loop.

## 6. Conclusions

The proposed SCCPP algorithm is the online algorithm that generates a traversable collision-free trajectory based on clothoids with low computational cost. Such a path is suitable and feasible for nonholonomic mobile robots since it does not contain sharp turns. By using a smoothing technique on the proposed coverage path, the coverage efficiency can be significantly improved in terms of the time required and energy consumption during the coverage tasks and has very low overlap redundancy. The complexity of the environment affects the coverage efficiency, and the experimental results evaluated the efficiency of the CCPP algorithms on maps with different complexity levels. By using the SCCPP algorithm, the trajectory followed by the robot can be executed faster and with higher accuracy than without the smoothing algorithm. The SCCPP algorithm produces the shortest coverage path, takes the shortest time for coverage execution, and has the smallest coverage redundancy compared to the CCD* and HDCP algorithms. The SCCPP algorithm takes advantage of the large size of the grid cells to ensure real-time operation and minimization of overlapping areas, but at the cost of a lower coverage rate due to the uncovered areas around obstacles and walls. However, the coverage rate could be easily increased by simply combining our SCCPP algorithm with a wall following algorithm.

As future work, more experiments are planned for other robot designs such as omnidirectional mobile robots and Ackermann steering vehicles. Furthermore, we consider the extension of this work to multiple robots in the form of a decentralized solution for the coordinated multi-robot complete coverage task. This will decrease the total task time significantly due to the division of workload overall robots, while decentralization will prevent a single point of failure.

## Figures and Tables

**Figure 1 sensors-22-09269-f001:**
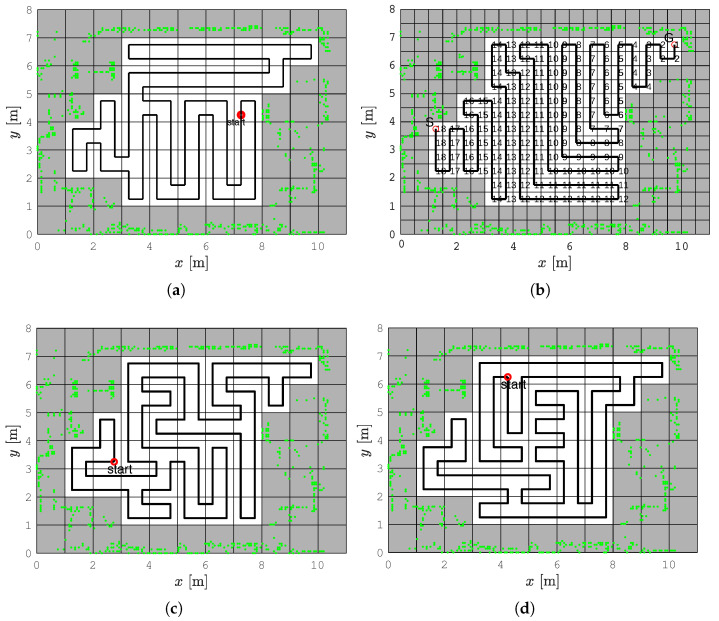
Solution examples of different methods for the TSP problem. (**a**) The RSTC algorithm, (**b**) The PT algorithm, (**c**) The ACO algorithm, (**d**) The HNN algorithm.

**Figure 2 sensors-22-09269-f002:**
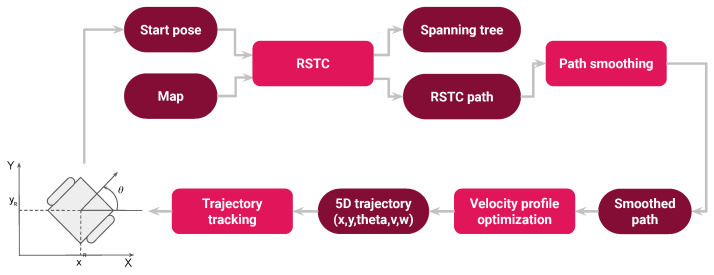
The pipeline of the smooth complete coverage path planning algorithm.

**Figure 3 sensors-22-09269-f003:**
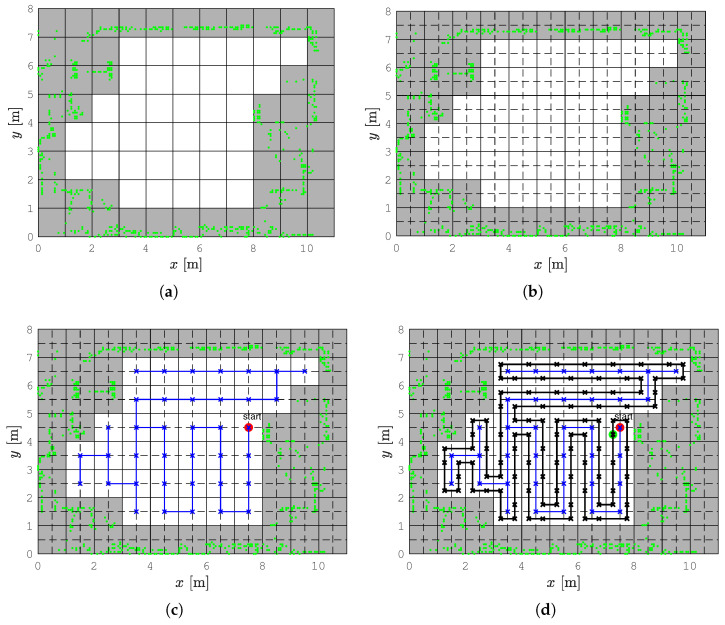
Example of the complete coverage path planning for the known environment step-by-step. The real obstacles collected by SLAM algorithm are presented with green dots. (**a**) Low resolution occupancy grid map, (**b**) Subdividing of grid cells, (**c**) The spanning tree (blue line), (**d**) Complete coverage path (black line).

**Figure 4 sensors-22-09269-f004:**
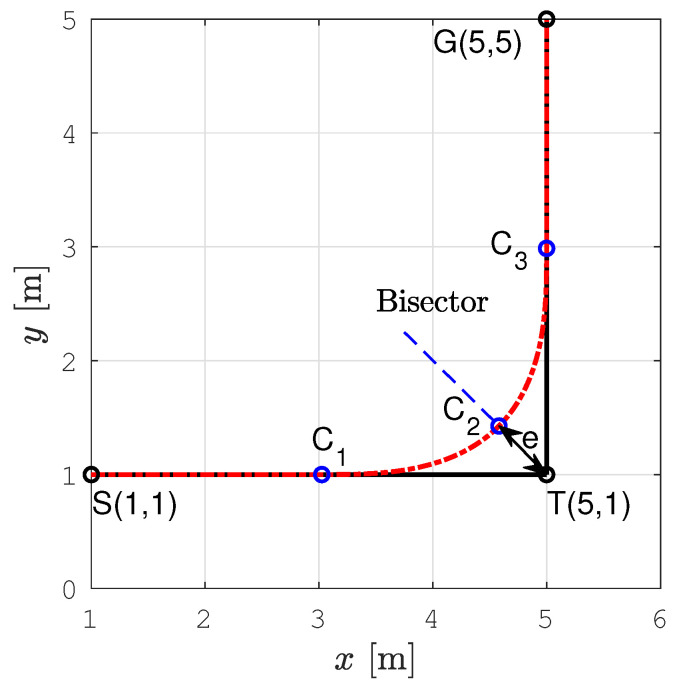
Illustration of the sharp turn smoothing by two clothoids.

**Figure 5 sensors-22-09269-f005:**
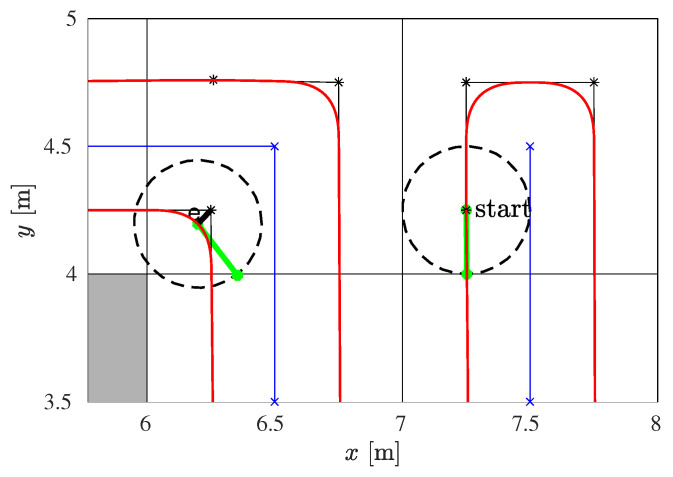
The enlarged part of the coverage path with two positions of the robot: (1) the robot’s starting position and (2) the robot’s position at the obstacle corner. The obstacle cell is shown in gray, the robot’s positions in dashed circles, the robot’s orientation in green, the spanning tree in blue, the RSTC path in black, and the smoothed path in red.

**Figure 6 sensors-22-09269-f006:**
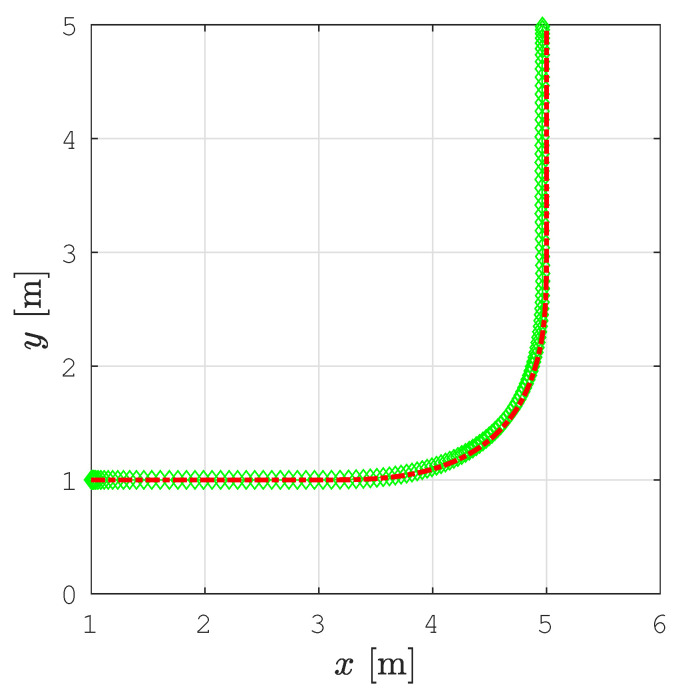
The smoothed path from Figure 4 (red line) and the tracked trajectory (green diamonds).

**Figure 7 sensors-22-09269-f007:**
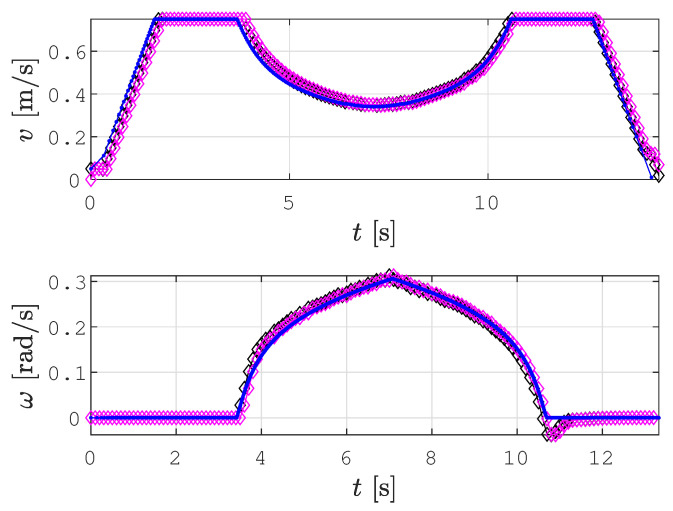
The calculated linear and angular velocities (blue line) and real linear and angular velocities (magenta diamonds) at which the robot tracked the trajectory.

**Figure 8 sensors-22-09269-f008:**
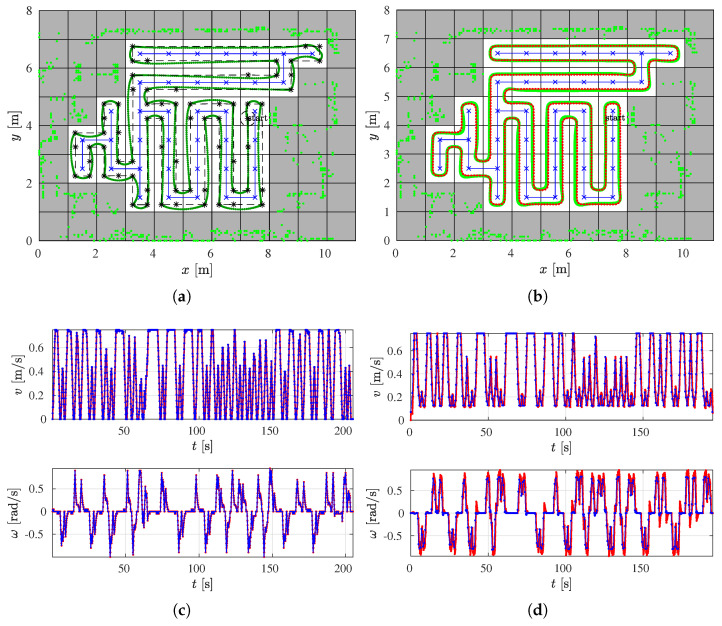
The comparison of the CCPP and SCCPP algorithms in the *Lab scenario* with the presented RSTC spanning tree (blue line), the RSTC path (black line), the smoothed path by the SCCPP algorithm (red line), the executed path by the CCPP and SCCPP algorithm (green line), the reference velocity profile (blue line), and the actual velocity profile (red line). (**a**) The CCPP path in the *Lab scenario*. (**b**) The SCCPP path in the *Lab scenario*. (**c**) The CCPP velocity profile in the *Lab scenario*. (**d**) The SCCPP velocity profile in the *Lab scenario*.

**Figure 9 sensors-22-09269-f009:**
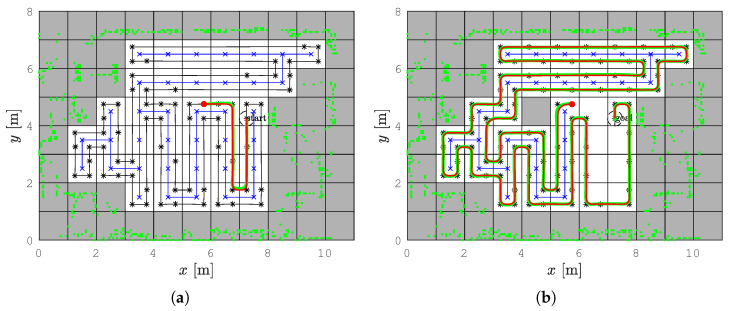
The RSTC algorithm in the *Lab scenario* with the presented RSTC spanning tree (blue line), the RSTC path (black line), the smoothed path by the SCCPP algorithm (red line), and the executed path by the smoothed RSTC algorithm (green line). Red points are the robots’ position where the replanning algorithm is executed. (**a**) The smoothed RSTC path in the *Lab scenario* before replanning. (**b**) The smoothed RSTC path in the *Lab scenario* after replanning.

**Figure 10 sensors-22-09269-f010:**
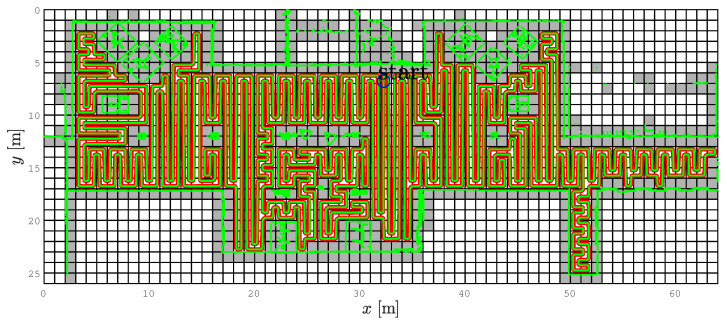
The SCCPP algorithm in the *Aula scenario*, which presents the smoothed path with red line and the executed path with green line.

**Figure 11 sensors-22-09269-f011:**
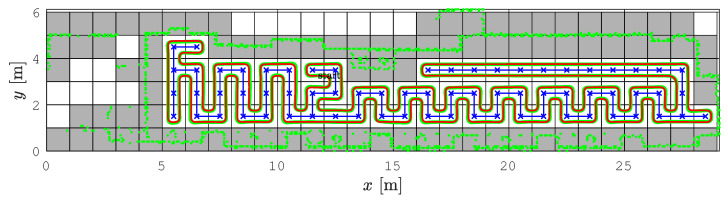
The SCCPP algorithm in the *Gallery scenario*: the RSTC spanning tree (blue line), the smoothed path (red line), and the executed path (green line).

**Figure 12 sensors-22-09269-f012:**
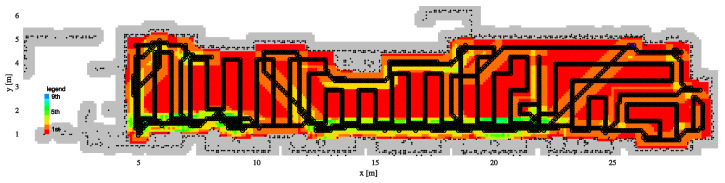
The complete coverage of the CCD* algorithm with the path and noted redundant numbers of cell visits (from 1 to 9).

**Figure 13 sensors-22-09269-f013:**
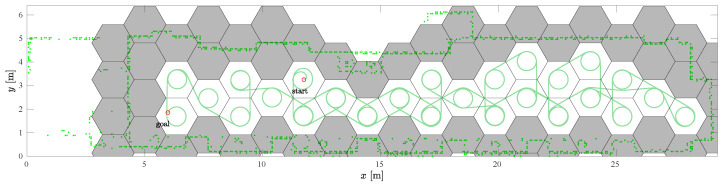
The complete coverage of the HDCP algorithm with hexagonal cell grid.

**Figure 14 sensors-22-09269-f014:**
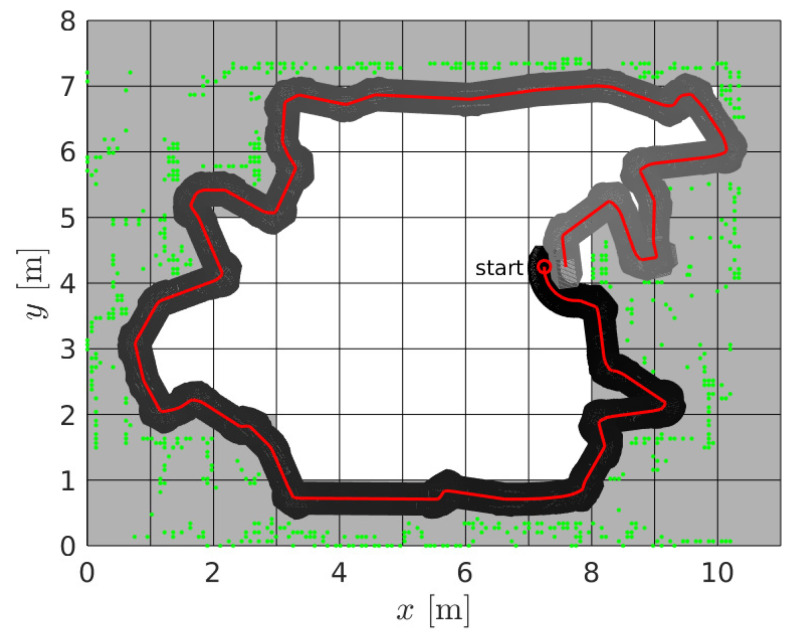
The wall following algorithm after SCCPP: driven trajectory (red line), and robot’s footprint in every point on the trajectory is presented degraded from black to gray.

**Figure 15 sensors-22-09269-f015:**
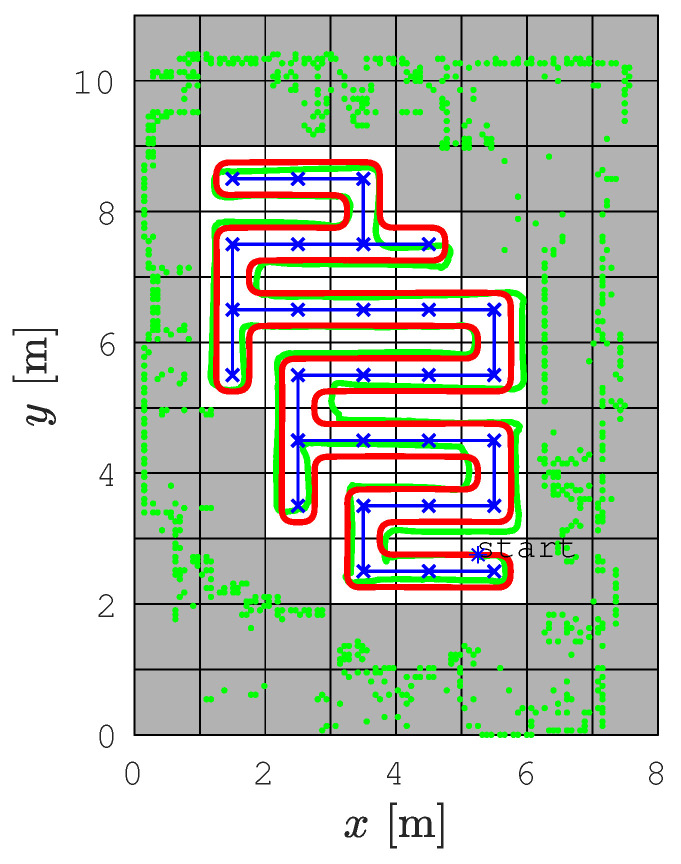
The calculated SCCPP path (red line) and trajectory tracked by the Pioneer 3DX robot (green line).

**Figure 16 sensors-22-09269-f016:**
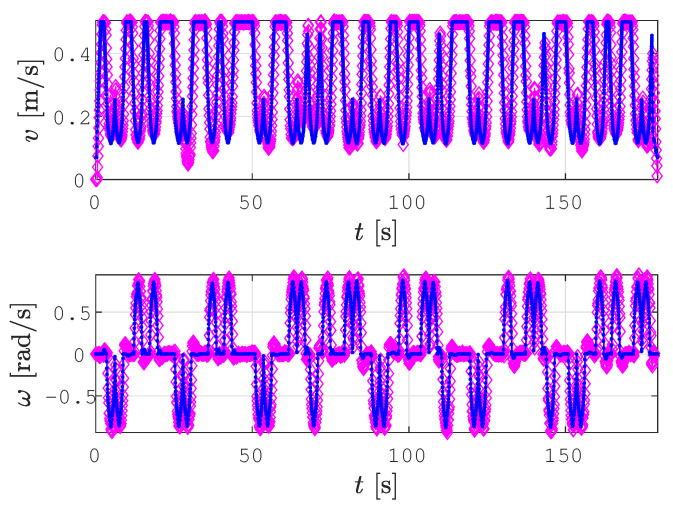
Calculated linear and angular velocities (blue line) and linear and angular velocities, while the Pioneer 3DX robot is tracking the trajectory (magenta line).

**Figure 17 sensors-22-09269-f017:**
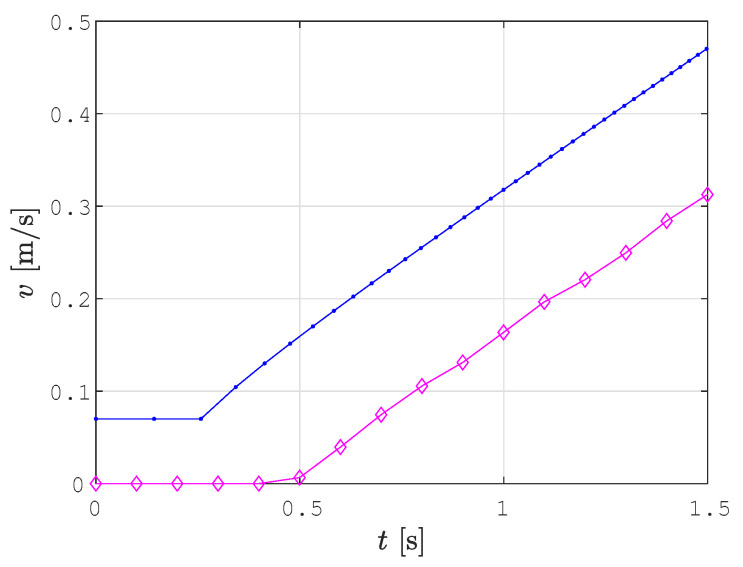
Real linear velocity, while the robot Pioneer 3DX is following the trajectory (magenta line), deviates from the applied linear velocity (blue line).

**Table 1 sensors-22-09269-t001:** Table of comparison of CCPP and SCCPP algorithms.

	Lab CCPP	Lab SCCPP	Gallery CCPP	Gallery SCCPP	Aula CCPP	Aula SCCPP
Coverage path length	77.22 m	**72.46 m**	141.04 m	**129.44 m**	1255.91 m	**1191.22 m**
Coverage time	217.30 s	**193.26 s**	453.99 s	**409.58 s**	3479.51 s	**2834.23 s**
Coverage rate	71.48%	**74.42%**	66.37%	**72.11%**	79.03%	**79.84%**
Coverage redundancy	35.53%	**7.89%**	48.19%	**5.79%**	39.23%	**3.37%**
Nodes number	61	61	97	97	744	744
Path calculation time	**0.5 ms**	0.9 ms	**0.6 ms**	1.2 ms	**5 ms**	10 ms

**Table 2 sensors-22-09269-t002:** Comparisons of the SCCPP, CCD* and HDCP algorithms for the *Gallery scenario*.

	SCCPP	CCD*	HDCP
Coverage path length	**129.44 m**	222.79 m	163.48 m
Coverage time	**409.58 s**	768.22 s	879.65 s
Coverage rate	72.11%	**98.85%**	57.38%
Coverage redundancy	**5.79%**	87.48%	46.15%

**Table 3 sensors-22-09269-t003:** SCCPP algorithm on the Pioneer 3DX robot for the *Lab scenario*.

	SCCPP
Coverage path length	53.85 m
Coverage time	179.5 s
Coverage rate	77.7%
Coverage redundancy	12.96%
Tracking error	6.43 $m^{2}$

## Data Availability

Not applicable.

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
