# Peer review of "Smooth Complete Coverage Trajectory Planning Algorithm for a Nonholonomic Robot"

_sensors, 2022, doi:10.3390/s22239269_

Round 1
Reviewer 1 Report
The paper presents a CCPP algorithm which generates smooth complete coverage paths that allow nonholonomic mobile robots to move in optimal time. My comments:
1.- The nature of this work is interesting
2.- However, the section 5 only present very little information. The analysis is highly insufficient, and lacks of scientific/experimental evaluation.
3. No comparison presented in the paper. Is there any comparison made to the existing method?
Reviewer 2 Report
In this paper, a planning algorithm based on stereo algorithm to generate smooth complete coverage path is proposed, which allows nonholonomic mobile robots to move along the path, which is a very interesting and attractive algorithm.
The following are some issues which are deserved the authors to concern on.
1) Whether the pseudo code of the algorithm can be provided for reading.
2) how about robustness of the proposed method?
3) In the dynamic situation, such as the sudden fire in the building and other special situations, whether the algorithm is effective, and whether the robot can move without collision.
4)The authors may make more comparisons with similar methods, such as the methods in the paper An Approximation Algorithm for Graph Partitioning via Deterministic Annealing Neural Network;Solving the Production Transportation Problem via a Deterministic Annealing Neural Network Method.
5) A complex figure is given in Section 4.2, which deserves more explanation.
6) Some sentences should be polished carefully in order to improve the readability.
